# Hypoglycemic events and glycemic control effects between NPH and premixed insulin in patients with type 2 diabetes mellitus: A real-world experience at a comprehensive specialized hospital in Ethiopia

**Ashenafi Kibret Sendekie** *, **Adeladlew Kassie Netere, Eyayaw Ashete Belachew**

Department of Clinical Pharmacy, School of Pharmacy, College of Medicine and Health Science, University of Gondar, Gondar, Ethiopia

* ashukib02@yahoo.com, Ashenafi.kibret@uog.edu.et

**Data Availability Statement:** All relevant data are within the manuscript and its Supporting Information files.

## Abstract

### Background

Though initiation of insulin results in a significant change in glycemic levels, treating patients without significant hypoglycemic events remains difficult in diabetes patients initiated with different insulin-based regimens. This study assessed the association of hypoglycemic incidence and glycemic control between NPH and premixed insulin regimens in patients with type 2 diabetes mellitus (T2DM).

### Methods

This was a retrospective observational study in patients with T2DM who were treated with insulin-based therapy from 2015 to 2020 at the University of Gondar Comprehensive Specialized hospital. Average fasting blood glucose (FBG) between NPH and premixed insulin regimens was compared using an independent t-test. The Association of NPH and premixed insulin regimens with hypoglycemic incidences and glycemic control was examined by a logistic regression model. P < 0.05 was statistically significant.

### Results

From 405 participants, more than half (55.3%) were males with a mean age of 59.2(±9.1) years. Baseline mean HbA1C and FBG levels were 12.73(±1.1) % and 347.7(±48.5) mg/dl, respectively. Within a one-year follow-up period of insulin initiation, the rate of hypoglycemia was 13.1%. The incidence of hypoglycemia was significantly higher in patients initiated with premixed insulin compared with NPH insulin regimens (P < 0.001). After one year of insulin initiation, HbA1C decreased from 12.7 to 7.6 and from 12.8 to 7.3% and FBG levels decreased from 347.5 to 160.7 and from 348.2 to 147.3 mg/dl following initiation of NPH and premixed insulin, respectively. Patients treated with premixed-based insulin were found

**Funding:** The authors received no specific funding for this work.

**Competing interests:** The authors have declared that no competing interests exist.

**Abbreviations:** ADA, American diabetes association; FBG, fasting blood glucose; HbA1C, glycosylated hemoglobin; NPH, Neutral Protamine Hagedorn; OADs, Oral Antidiabetics; T2DM, Type 2 Diabetes Mellitus.

more likely to achieve target FBG compared with patients treated with NPH insulin regimens after one year of initiation (P = 0.02).

## Conclusion

Premixed insulin-based regimen has found to have a higher hypoglycemic incidence, but a better level of glycemic control compared to NPH insulin-based therapy. Therefore, patients initiated with premixed insulin need to be highly vigilant and motivated to recognize the symptoms of hypoglycemia.

## Introduction

Because of an increase in obesity, physical inactivity and sedentary lifestyle in developing countries, the prevalence of type 2 diabetes mellitus (T2DM) has increased rapidly worldwide [1]. It accounts for more than 95% of diabetes cases [1]. These patients exposed to T2DM for a longer time may become insulin deficient, and controlling hyperglycemic states with lifestyle modifications and oral antidiabetic medications alone could be impossible. Consequently, because of the progressive nature of the disease, patients with type 2 diabetes mellitus (T2DM) have been initiated with insulin-based therapy [2–4]. Despite recommendations from guidelines, real-world practices regarding insulin treatment vary widely across the countries. Patients with T2DM and long-lasting diseases ultimately require treatment intensification with insulin alone or along with oral antidiabetic agents. Clinicians would consider insulin initiation for patients who remain above the target glucose level despite maximal oral antidiabetic and/or other non-insulin injectable medications for a duration of 3–6 months [3,5–7].

Though more intensified insulin regimens have a potential benefit in the lowering of blood glucose and delaying the onset of complications, they may also have negative possible effects like higher risk of weight gain, hypoglycemia and hyperinsulinemia [8]. Either basal or prandial insulin therapy could be initiated depending on the patients' lifestyle patterns and needs, and the regular practice of the clinicians. At the initiation of therapy, initial regimens may help patients achieve blood glucose targets [7,9,10], but because of the progressive nature of the disease, finally both basal and prandial insulin will be required to maintain glycemic level within the target range [11]. An observational study in five counties showed that physicians from Germany often treat newly diagnosed T2DM patients with a short acting early intensive insulin initiation than favoring oral antidiabetic agents [12]. But in Spain and France, long or/and intermediate acting insulin regimens were the most commonly used regimens, whereas long and/or intermediate-acting or premixed insulin preparations were commonly used in the United Kingdom and Greek [12]. Finally, this study has found a significant change in blood glucose levels without significant risk of self-reported hypoglycemia.

Although the initiation of different insulin regimens has a significant effect on glycemic control and delays the onset of complications [13–15], treating patients without a significant risk of hypoglycemia in the real-world practices remains difficult. Hypoglycemia results from a mismatch between insulin and carbohydrate intake, exercise, or alcohol consumption. Concerns about the risk of hypoglycemia can prevent or delay the initiation or intensification of insulin therapy [16]. Of patients taking insulin, 7%–15% experience at least one episode of hypoglycemia per year, and 1%–2% found to have severe hypoglycemia (i.e., requiring assistance from others for treatment), and hypoglycemia has been associated with poor outcomes and higher rates of death [17–19]. Patients with T2DM and a history of at least one severe

hypoglycemic event have an approximately two-to-four-fold higher death rate than those who have not had a severe event [20–22]. Therefore, while choosing and initiating insulin regimens, it is important to consider the need of treatment intensification and its benefits over the potential risks [23]. All patients should be educated about the symptoms and self-treatment of hypoglycemia. The American Diabetes Association (ADA) recommends checking of the blood glucose level if signs or symptoms of hypoglycemia are present. Additionally, if the blood glucose level is less than 70 mg per treat, patients could take fast-acting carbohydrates like fruit juice or three or four glucose tablets, and recheck the blood glucose level after some minutes to ensure that it has normalized [10].

Advanced settings had a real-world demonstration regarding hypoglycemia incidences associated with the initiation of different regimens of insulin, such as basal and short acting insulin preparations [24–27]. However, there is insufficient real-world data in the resource-limited settings, particularly in the study area. Two types of insulin regimens such as NPH and premixed (regular 30% plus NPH 70%) insulin, have been initiated for the outpatient management of patients with T2DM having follow-up at the Ethiopian hospitals. However, the association of hypoglycemic episodes and glycemic control effects with these insulin regimens has not yet investigated so far. Therefore, this study assessed the rate of hypoglycemia incidence and level of glycemic control between NPH and premixed insulin regimens among patients with T2DM at the University of Gondar Comprehensive Specialized hospital (UoGCSH).

## Methods and materials

### Study design, setting and participants

A retrospective follow-up study was conducted at UoGCSH from 2015 to 2020 using medical records of NPH and premixed insulin-initiated patients with T2DM. UoGCSH is among one of the largest comprehensive and specialized hospitals in Ethiopia and serves for around 9 million population in the catchment of Gondar, including special care of diabetes patients. Adult patients aged 18 and above who initiated either NPH or premixed insulin after inability to achieve target blood glucose level by oral antidiabetic agents alone were enrolled and then followed for a one-year post-insulin initiation period. Patients with T2DM who were treated with insulin-based therapy (NPH and premixed) during the indexed period of 2015 to 2020 were included in the study. Participants should have a one-year data starting from the index date (the date at which insulin was prescribed for the first time). Participants were excluded in the study if they had altered insulin, received gestational or type 1 diabetes, and had incomplete medical records throughout the study period.

### Sampling size and sampling techniques

The sample size was computed using a single population proportion formula: n = Z2 p (1-p)/ W, n = required sample size, Z = the degree of accuracy required (95% level of significance = 1.96), p = the rate of hypoglycemic incidences and/or rate of glycemic control of T2DM patients on either NPH or premixed insulin assumed to be 50% (0.5), it is to obtain maximum sample size, W = marginal error of 5% (0.05). By considering a 10% of possible incomplete medical records of the participants, we enrolled to approach 423 participates. Then, a 2:1 ratio of participants, 282 vs 141 who were treated by NPH and premixed insulin-based therapy, respectively, were allocated to the final study. This is because the trend in the patients' record showed that few patients were treated with premixed insulin-based therapy compared with NPH insulin.

Eligible samples were included in the study using a systematic random sampling technique. The first sample used as the starting point was selected using a simple random technique using

a lottery method. Then, the study participants were included using the sampling interval until an adequate sample size was achieved.

## Data collection instruments and procedures

The data extraction tool was prepared from previous literature with modifications by considering the setting and nature of the medical records of the participants. The data extraction tool consisted of socio-demographic, clinical, and medication characteristics of the participants. Socio-demographic characteristics include; patents' sex, age, residency, and duration of T2DM since diagnosis. The clinical characteristic section consists of physician and nurse notes, various laboratory results, diagnoses and related comorbidities and complications. Whereas medications include the type of antidiabetics medications, antihypertensive agents, lipid-lowering agents and other groups of medications used to treatment of the existed comorbidities and complications.

The data collectors approached the eligible participants using their identification medical cards. From the indexed date, one-year data was recorded every three months from stored physical medical records, and printed laboratory results were checked for some laboratory tests. For instance, levels of FBG, HbA1C, serum creatinine (Scr.), total cholesterol (TCl), and triglycerides (TGs) were checked from printed laboratory results. Participants were categorized to NPH and premixed insulin regimens at the indexed data based on the respective initiated insulin regimens, and the data was followed for one year of follow-up period. Insulin (NPH or premixed) was initiated after the patients could not achieve target blood glucose level with oral antidiabetic agents alone (Metformin alone or along with glibenclamide). Insulin initiation and intensification, including dosing titration and adjustment, were applied based on the ADA recommendations.

## Operational definitions

**Hypoglycemic unawareness (HU).**   It is a reduction in the ability to recognize low blood glucose levels in patients with diabetes, and might lead to potentially lethal consequences in patients with comorbidities. Patients with autonomic neuropathy may have HU, but patients don't feel the symptoms of low blood glucose.

## Data quality management, entry and statistical analysis

Before the actual data collection, the data collectors and the supervisor were trained about the data extraction techniques. All were clear for the objective and ethical aspects of the study. Pretest was performed on 5% of the sample size and some modification of the tool was applied. After the medical record identification numbers of the participants were entered to the Microsoft 2016, the repetition was checked and the data extraction was conducted. The data collectors and the supervisor checked the data quality and completeness after each day of the data collection. The supervisor explicitly followed the data collection closely. After assuring the data completeness, cleanness and quality, it was coded and entered into Epi-Info version 8 and exported to Statistical Package for Social Sciences (SPSS) software version 26 for statistical analysis.

In the descriptive statistics, categorical variables were presented using frequencies and percentages, while the mean with standard division was used to present continuous variables. The normal distribution of the data was explored using Q-Q plot, histogram and the Shapiro–Wilk test. An independent t-test was used to examine the difference in the level of blood glucose between NPH and premixed initiated participants. A logistic regression model was applied to identify the association between insulin regimens and other variables with the incidence of

hypoglycemia and glycemic control. Variables with $P \leq 0.2$ in bivariable analysis were entered to the multivariable model to identify the potential associated factors. A P-value $< 0.05$ at 95% confidence interval (CI) was statistically significant.

### Outcome measures

Hypoglycemic incidence and glycemic control effects between NPH and premixed insulin among patients with T2DM treated with insulin regimens are the outcome of this study. Then, the rate and association of the hypoglycemic incidence and glycemic control effects were compared between these two regimens.

**Hypoglycemia.** It was measured using a clinical episode of hypoglycemia and/or recorded blood glucose level $< 70$ mg/dl on the patients' medical record within the one-year insulin-initiation period. Patients who had at least one record of a clinical episode of hypoglycemia or a blood glucose record $< 70$ mg/dl were taken as having hypoglycemia. Then, the incidence was measured and reported using the number of hypoglycemic episodes during a one-year follow-up period per total participant. Hypoglycemia was then classified as non-sever and sever based on the patients' clinical condition and need of admission to the emergency ward, and level of recorded blood glucose. Severe hypoglycemia was considered in patients who sought emergency treatment or/and in patients with a record of blood glucose $< 54$ mg/dl [28] over a one-year follow-up period.

**Glycemic control.** From the time of insulin-initiation (NPH versus premixed), a one-year follow-up level of blood glucose was recorded every three months. Then, the average glycemic level was computed, and the glycemic control was measured using the level of average HbA1c categorized as poor; HbA1C $\geq 7\%$ and good; HbA1C level $< 7\%$ for adult patients [29].

### Ethical considerations

The proposal was ethically approved by the ethical review committee of the University of Gondar (UoG), reference number Sop/037/2020. Informed consent of the participants was not applicable and it was waived by ethical review committee of UoG. The privacy and confidentiality of the participants were maintained and the data was adequately anonymized.

## Results

### Socio-demographic and baseline clinical characteristics

Of the 423 enrolled participants, 405 samples were included in the study. More than half (55.5%) of the participants were males with a mean ($\pm$SD) age of 59.2($\pm$9.1) years. In addition to T2DM, most of the participants had hypertension (67.2%) followed by dyslipidemia (35.1%). The mean HbA1C level at the indexed time was 12.73($\pm$1.1) % (**Table 1**).

### Medications used in the study participants

Following insulin initiation, most of the study participants (59.5%) received a combination of metformin and insulin. Regarding the type of insulin, more than two-thirds (67.9%) of the study participants were on NPH insulin-based therapy. Enalapril (59%) and atorvastatin (34.6%) were the most prescribed antihypertensive and lipid-lowering agents, respectively. The average ($\pm$SD) daily dose of insulin at the time of initiation was 16.8 ($\pm$5.9) and 27.7($\pm$8.5) mg at the end of follow-up period (**Table 2**).

**Table 1. Socio-demographic and baseline clinical characteristics of inulin-initiated T2DM patients at UoGCSH from 2015 to 2020 (N = 405).**

| Socio-demographic variables | | Frequency (%) | Mean (±SD) |
|---|---|---|---|
| Sex | Male | 224 (55.3) | |
| | Female | 181(44.7) | |
| Age in years | | - | 59.2(±9.1) |
| Weight in Kg. | | - | 65.6(±8.3) |
| Years since T2DM diagnosis | | - | 13.3(±3.9) |
| Residence | Urban | 242(59.8) | |
| | Rural | 163(40.2) | |
| **Clinical characteristics** | | | |
| Hypertension | | 272(67.2) | |
| Dyslipidemia | | 142 (35.1) | |
| Macrovascular complications | | 61 (15.1) | |
| Microvascular complications | | 25 (6.2) | |
| Diabetes ketoacidosis | | 21 (5.2) | |
| Renal disorders | | 14(3.5) | |
| Others* | | 14(3.5) | |
| Indexed time average HbA1C (%) | | | 12.73(±1.1) |
| Indexed time average FBG (mg/dl) | | | 347.7(±48.5) |
| Systolic blood pressure (mmHG) | | - | 137.2(±11.7) |
| Diastolic blood pressure (mmHG) | | - | 81.2(±9.5) |
| Serum creatinine (mg/dl) | | - | 1.9(±0.9) |
| Total cholesterol (mg/dl) | | - | 196.0(±49.6) |
| Total glyceride (mg/dl) | | - | 168.6(±45.6) |

FBG, Fasting blood glucose; Others*, Thyrotoxicosis, bronchial asthma, retroviral infections, skin disorders, bacterial infections.

## Comparison of glycemic level between NPH and premixed insulin regimens

Glycemic levels were compared between NPH and premixed insulin-treated patients. On average, the study participants who initiated premixed insulin had a little bit worse; FBG (Mn = 348.2) and HbA1C (Mn = 12.8) levels than patients who initiated NPH insulin; FBG (Mn = 347.5) and HbA1C (Mn = 12.7) at the index date. The difference was not statistically significant. However, there was a statistically significant difference in the level of blood glucose after insulin initiation with better FBG and HbA1C levels among patients treated with premixed insulin compared with NPH insulin. Overall, the average FBG after one-year insulin initiation was 156.4 mg/dl (ranges: 80–247) and the average HbA1C was 7.5% (ranges: 5.3–10). Furthermore, the average FBG level decreased from 347.5 to 160.7 mg/dl after one-year initiation of NPH insulin and from 348.2 to 147.3 mg/dl after one-year initiation of premixed insulin (**Table 3** and **Fig 1**). While the average HbA1C was decreased from 12.7 to 7.6% after one-year initiation of NPH insulin and from 12.8 to 7.3% after one-year initiation of premixed insulin (**Table 3**). Moreover, in one year of insulin initiation, the overall average time to achieve target blood glucose was 6.6 months. Compared with NPH insulin (Mn = 7.1 months), time to achieve target blood glucose was significantly shorter in patients who were initiated with premixed insulin (Mn = 5.9 months) within a one-year follow-up period (p = 0.013).

## Comparison of hypoglycemic episodes and glycemic control effects

Overall, 53 (13.1%) participants experienced at least one hypoglycemic episode, and 9 (2.2%) were attacked by sever hypoglycemia. Patients who were treated with premixed-based insulin regimen were found more likely to have hypoglycemia (34.6%) compared with those treated

**Table 2. The distribution of medications in NPH and premixed insulin-initiated T2DM patients.**

| Medications | Category | Frequency (%) | Mean (SD) |
|---|---|---|---|
| Antidiabetic regimens | Metformin plus insulin | 241(59.5) | |
| | Metformin plus glibenclamide plus insulin | 140 (34.6) | |
| | Insulin alone | 24 (5.9) | |
| Type of insulin | NPH | 275 (67.9) | |
| | Premixed | 130 (32.1) | |
| Antihypertensive agents | Enalapril | 239 (59) | |
| | Hydrochlorothiazide | 72 (17.8) | |
| | Amlodipine | 23 (5.7) | |
| | Furosemide | 18 (4.4) | |
| | Atenolol | 15 (3.7) | |
| | Metoprolol | 14 (3.5) | |
| | Nifedipine | 14(3.5) | |
| Lipid–lowering agents | Atorvastatin | 140 (34.6) | |
| | Simvastatin | 48 (11.9) | |
| Aspirin | - | 68 (16.8) | |
| Others* | - | 14 (3.5) | |
| Average daily dose of insulin | At the start of initiation | | 16.8(±5.9) |
| | At the end of follow-up | | 27.7(±8.5) |
| Average daily dose of lipid-lowering agents | Atorvastatin | | 41.4(±10.4) |
| | Simvastatin | | 22.2(±6.4) |

Others

*; antiretroviral, antibiotics, propyl thiouracil, beclomethasone, amitriptyline.

by NPH-based insulin regimen (2.9%) (p < 0.001). After one-year of insulin initiation, the overall rate of good glycemic control was 24.9%. Compared with NPH-based insulin (18.9%), a significantly greater proportion of patients treated by premixed-based insulin (31.5%) had achieved glycemic control (p < 0.001) (**Table 4**).

## Association of NPH and premixed insulin and other variables with hypoglycemia

The adjusted multivariable logistic regression analysis showed that premixed insulin-based regimen and microvascular compilation were significantly associated with hypoglycemia

**Table 3. Comparison of blood glucose levels between NPH and premixed insulin regimens.**

| Follow-up times | Mean (±SD) blood glucose level | | P-value |
|---|---|---|---|
| | NPH | Premixed | |
| Baseline HbA1C | 12.7(±1.2) | 12.8(±1.1) | 0.59 |
| Baseline FBG | 347.5(±49.3) | 348.2 (±46.9) | 0.89 |
| 3rd month FBG | 184.4(±48.6) | 163.8(±55.8) | < 0.001* |
| 6th month FBG | 166.9(±43.4) | 153.8(±49.2) | 0.01* |
| 9th month FBG | 149.7(±41.6) | 138.3(±34.9) | 0.004* |
| 12th month FBG | 141.7(±38.7) | 133.3(±30.4) | 0.018* |
| Average FBG after one year | 160.7(±32.4) | 147.3(±32.4) | < 0.001* |
| Average HbA1C after one year | 7.6(±0.8) | 7.3(±0.8) | < 0.001* |

FBG is measured in mg/dl, and HbA1C values with %.

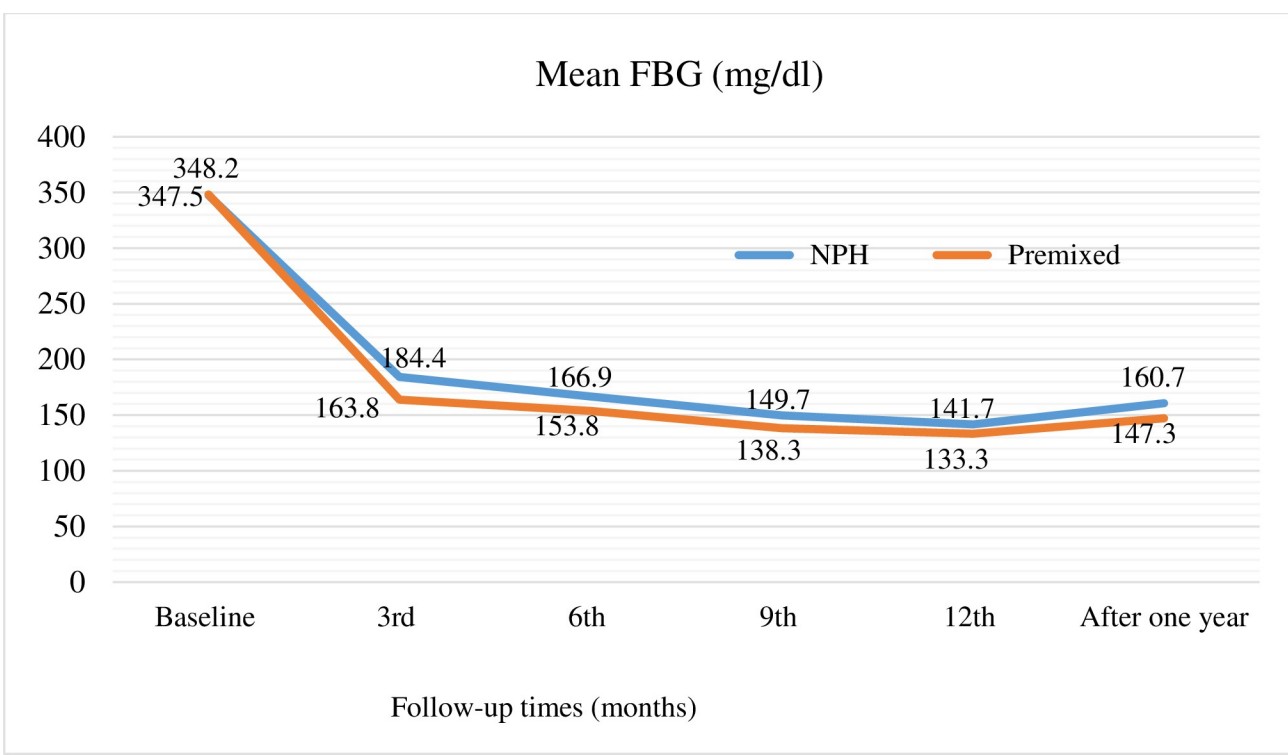

**Fig 1. Blood glucose levels during a one-year follow-up period of inulin (NPH vs premixed) initiation.**

events. Patients who were treated with a premixed insulin-based regimen were found 96.7% more likely to have hypoglycemia than patients initiated with NPH insulin [AOR = 0.033, 95% CI (0.009–0.119); p < 0.001]. Similarly, patients who had microvascular complications were found 80.4% more likely to have hypoglycemia compared with patients who were without microvascular complications [AOR = 0.196, 95% CI (0.04–0.968); p = 0.046] (**Table 5**).

## Association of NPH and premixed insulin and other variables with glycemic control

After controlling of other variables, the multivariable analysis showed that premixed insulin-based regimen and lipid-lowering agents were found significantly associated with glycemic control. Patients who were initiated with a premixed insulin-based regimen were found less likely to have poor glycemic control than patients treated with NPH insulin [AOR = 0.421, 95% CI (0.065–0.973); p = 0.02]. In contrast, patients who received atorvastatin were found

**Table 4. The rate of hypoglycemia and glycemic control between NPH and premixed insulin regimens.**

| Variables | | Type of insulin | | Overall rates | P-value |
|---|---|---|---|---|---|
| | | NPH (N = 275) | Premixed (N = 130) | | |
| Hypoglycemia during a one-year follow-up period of insulin initiation | Yes | 8 (2.9%) | 45(34.6%) | 53(13.1%) | < 0.001* |
| | No | 297(97.1%) | 85(65.4%) | 253(86.9%) | |
| Overall rate of glycemic control after one year of insulin initiation | Good | 52(18.9%) | 49(37.7%) | 101(24.9%) | < 0.001* |
| | Poor | 223(81.1%) | 81(62.3%) | 304(75.1%) | |

* Indicates p < 0.05.

**Table 5. Association of NPH and premixed insulin and other predicted variables with hypoglycemia.**

| Variables | | Hypoglycemia | | 95% CI | | P-value |
|---|---|---|---|---|---|---|
| | | No | Yes | COR | AOR | |
| Sex | Male | 190 | 34 | 0.655(0.36–1.193) | 0.609(0.202–1.836) | 0.378 |
| | Female | 162 | 19 | 1 | 1 | |
| Residency | Urban | 209 | 33 | 0.886(0.489–1.606) | 0.526(1.55–1.783) | 0.302 |
| | Rural | 143 | 20 | 1 | 1 | |
| Dyslipidemia | Yes | 118 | 24 | 0.609 (0.34–1.903) | 0.171(0.029–1.028) | 0.054 |
| | No | 234 | 29 | 1 | 1 | |
| Macrovascular complication | Yes | 50 | 11 | 0.632(0.305–1.309) | 0.943(0.249–3.577) | 0.193 |
| | No | 302 | 42 | 1 | 1 | |
| Microvascular Complications | Yes | 19 | 6 | 0.447(0.170–1.176) | 0.196(0.04–0.968) | 0.046* |
| | No | 333 | 47 | 1 | 1 | |
| Antidiabetic mediations | Insulin plus metformin | 214 | 27 | 1.585(0.504–4.985) | 0.451(0.055–3.695) | 0.580 |
| | Insulin plus metformin plus glibenclamide | 118 | 22 | 1.073(0.334–3.443) | 0.753(0.084–6.718) | |
| | Insulin | 20 | 4 | 1 | 1 | |
| Types of insulin | Premixed | 85 | 45 | 0.057 (0.026–0.125) | 0.033(0.009–0.119) | < 0.001* |
| | NPH | 267 | 8 | 1 | 1 | |
| Lipid-lowering agents | Atorvastatin | 125 | 15 | 2.778(1.193–6.465) | 2.394(0.783–1.7.316) | 0.126 |
| | Simvastatin | 36 | 12 | 1 | 1 | |

AOR, adjusted odds ratio; COR, crude odds ratio; CI, confidence interval

* indicated p value < 0.05.

about 3 times more likely to have poor glycemic control compared with patients who received simvastatin [AOR = 2.934, 95% CI (1.249–6.894); p = 0.014] (**Table 6**).

## Discussion

To the best of the authors' literature search, there was no published article regarding hypoglycemic incidence and effects of glycemic control between NPH and premixed insulin regimens in the resource-limited settings, in particular to the study area. Thus, the current institutional-based retrospective follow-up study was designed to examine the rate of hypoglycemic incidence and level of glycemic control between NPH and premixed insulin regimens in patients with T2DM. although studies have shown that the initiation of insulin has a significant change in glycemic levels and positive clinical impacts [13–15,30–32], achieving target blood glucose levels without significant hypoglycemic events remains challenging in rea-world practice. This study examined the association of hypoglycemic incidence and glycemic control effects between NPH and premixed insulin regimens in insulin-initiated patients with T2DM. The study also assessed associated factors for hypoglycemic incidence and glycemic control.

The current study disclosed that following insulin initiation, a significant proportion of study participants experienced hypoglycemic episodes with an overall rate of 13.1% within a one-year of insulin-initiation period. The rate of sever hypoglycemia is reported to be 2.2%. This is consistent with the previous studies [17–19]. In line with the previous studies [8,15,27,33], there was also a significant difference in the incidence of hypoglycemic episodes including rate of sever hypoglycemia between NPH and premixed insulin regimens, which hypoglycemic episodes were more likely to occur in patients treated with premixed insulin-based regimens compared with patients treated with NPH insulin regimens. The finding might be justified that the regular insulin component of the premixed insulin is more likely to be responsible for hypoglycemic episodes because of its short acting effects. In contrast to this finding another earlier study showed that patients treated with premixed insulin experienced

**Table 6. Association of NPH and premixed insulin and other predicted variables with glycemic control.**

| Variables | | Glycemic control | | 95% CI | | P-value |
|---|---|---|---|---|---|---|
| | | Poor | Good | COR | AOR | |
| Sex | Male | 163 | 61 | 0.758(0.479.1.198) | 1.145(0.534–2.456) | 0.728 |
| | Female | 141 | 40 | 1 | 1 | |
| Residency | Urban | 180 | 62 | 0.913(0.576–1.448) | 1.236(0.532–2.87) | 0.622 |
| | Rural | 124 | 39 | 1 | 1 | |
| Average weight (Kg) | | 65.6(±8.2) | 65.6(±8.6) | 0.999(0.972–1.027) | 0.987(0.942–1.023) | 0.566 |
| Years since T2DM Dx. | | 13.5(4) | 12.7(3.6) | 1.059 (0.997–1.125) | 1.104–1.23) | 0.069 |
| Hypertension | Yes | 210 | 62 | 1.405(0.880–2.245) | 0.485(0.115–2.038) | 0.323 |
| | No | 94 | 39 | 1 | 1 | |
| Dyslipidemia | Yes | 102 | 40 | 0.770 (0.484–1.225) | 1.105(0.438–2.785) | 0.833 |
| | No | 202 | 61 | 1 | 1 | |
| Macrovascular complication | Yes | 39 | 22 | 0.528(0.296–0.944) | 0.526(0.215–1.282) | 0.157 |
| | No | 265 | 79 | 1 | 1 | |
| Hypoglycemia | Yes | 34 | 19 | 0.543(0.294–1.004) | 2.201(0.627–7.724) | 0.218 |
| | No | 270 | 82 | 1 | 1 | |
| Antidiabetic regimens at 12th month | Insulin plus metformin | 208 | 65 | 1.872(1.109–3.159) | 2.593(1.058–6.355) | 0.058 |
| | Insulin plus metformin plus glibenclamide | 43 | 5 | 5.03(1.802–14.043) | 6.173(1.056–36.07) | 0.063 |
| | Insulin | 53 | 31 | 1 | 1 | |
| Types of insulin | Premixed | 81 | 49 | 0.385 (0.242–0.614) | 0.421(0.065–0.973) | 0.020* |
| | NPH | 223 | 52 | 1 | 1 | |
| Lipid-lowering agents | Atorvastatin | 110 | 30 | 3.103(1.546–6.227) | 2.934(1.249–6.894) | 0.014* |
| | Simvastatin | 26 | 22 | 1 | 1 | |
| Enalapril | Yes | 187 | 52 | 1.506(0.957–2.370) | 1.269(0.35–4.596) | 0.717 |
| | No | 117 | 49 | 1 | 1 | |
| Amlodipine | Yes | 21 | 2 | 3.673(0.846–15.949) | 0.656(0.108–3.969) | 0.646 |
| | No | 283 | 99 | 1 | 1 | |

AOR, adjusted odds ratio; COR, crude odds ratio; CI; confidence interval

* indicated p value < 0.05.

fewer hypoglycemic episodes than NPH insulin [34]. However, in the previous study, the sample size was too small and the study was conducted in a short follow-up period. The findings may indicate that patients require strict follow-up and blood glucose monitoring to avoid the life-threatening effects of hypoglycemia. Therefore, patients who are highly capable of self-management would preferable to the initiation of premixed insulin regimens. Furthermore, patients initiated with a short acting insulin regimen as a component of premixed insulin may need to be highly vigilant and motivated to recognize the symptoms of hypoglycemia, and they required to be adherent with the medications and its administration techniques as well as the dosing.

This study also showed that patients with microvascular complications were found more likely to have hypoglycemia than patients without microvascular complications. This finding partially justifies those patients with microvascular complications might have developed hypoglycemia unawareness (HU) or impaired awareness of hypoglycemia (IAH) because of treatments aimed at strict glycemic control to delay the progression of these complications. Though hypoglycemia unawareness is less frequently in patients with T2DM, patients with autonomic neuropathy don't feel hypoglycemia. Previous studies also indicated that patients with neurological and cardiovascular complications are associated with HU [35,36]. Usually, patients do not experience or perceive the symptoms of hypoglycemia, which typically occur when the blood glucose levels fall below 54 mg/dl (3.0 mmol/L). Blood glucose monitoring, individualized targets and patient education are important to prevent hypoglycemia.

Therefore, patients would be aware of it, and healthcare providers could provide continues patient education.

In line with the previous studies [15,37–42], after a one year of insulin initiation, the overall rate of good glycemic control in patients with T2DM who couldn't achieve a target glycemic level with oral antidiabetic agents alone was around 25%. Regarding the association of glycemic control with the types of insulin regimens, consistently with the previous studies [8,15,27,34], patients who were treated with premixed insulin-based regimens were found to be more likely to have good glycemic control compared with patients those treated with NPH insulin. Patients initiated with premixed insulin have also achieved target glucose levels within a significantly shorter period than patients who were treated with NPH insulin-based regimen. Moreover, the change in blood glucose levels between the two regimens also had a significant difference in all follow-up periods of post insulin initiation. Patients who initiated a premixed insulin-based regimen were more likely to have better average glucose levels than patients who received NPH insulin-based therapy [24].

Studies indicate that patients who have been treated with the premixed insulin regimens have a better rate of good glycemic control. This might be because of the effect of the insulin preparations, which can adjust the level of glucose form both the basal insulin and the postprandial sources. Previous studies also revealed that the premixed insulin regimens are the most physiologic approach because of their effect on both the basal and prandial coverage [3,5–7,10]. However, the current study showed that despite a higher proportion of patients are under the level of poor glycemic control, still lower proportion of the participants are treated with premixed insulin therapy. Additionally, treatment intensification and dose titration were also limited. This might be because of the fear of side effects, such as hypoglycemia, and the need of strict follow-up and monitoring. Concerns about the risks of hypoglycemia can prevent or delay the intensification and titration of insulin [16]. Therefore, patients who are capable of self-monitoring blood glucose to avert the negative effects like hypoglycemia would be better to initiate with premixed insulin regimens. Additionally, the literature has suggested that the addition of short-acting insulin as a component of premixed insulin regimen is better in patients with an elevated post-prandial blood glucose level [43,44], which can adjust for meal-related insulin secretion defects. As a result, when choosing insulin for initiation, elevated postprandial glucose levels may guide treatment selection and identify patients in need of treatment intensification [45].

Consistent with the previous studies [15,46–48], this study showed that glycemic control was associated with lipid-lowering agents. Patients who were treated with atorvastatin were found more likely to have poor glycemic control compared with patients treated with simvastatin. Previous studies also demonstrated that high intensity dose of atorvastatin was associated with poor glycemic control compared with moderate intensity statins. This finding may indicate that statin therapy plays a role in the downregulation of glucose transport in adipocytes, which can result in insulin resistance and deterioration of glycemic levels in patients with diabetes, particularly with high intensity statin treatment. This study also showed that patients received high intensity dose of atorvastatin with an average daily dose of 41mg, which may contribute to the existing hyperglycemia.

Generally, this study highlights the extent of hypoglycemia and rate of glycemic control using real-world data among patients initiated with NPH and premixed insulin regimens, which is not demonstrated before in resource-limited settings, particularly in the study area. Therefore, it can be used as a bench mark for the coming studies, and the clinicians and patients may tailor their insulin-initiation based on the real-world data.

This study has some limitations. Since the study is a retrospective, some variables might be lost and not included in the presentation, for instance, hypoglycemia may be unaware for

patients and may not be reported and recorded in the medical records of the participants. Some variables like macro and microcomplications, may not be consistent throughout the follow-up period. Therefore, the results should be interpreted with caution. Prospective studies with larger populations would be recommended to better appreciate hypoglycemic episodes and glycemic control effects of different insulin regimens.

## Conclusion

The current study concluded that premixed insulin-based regimen found to have a high rate of hypoglycemic incidence with a significant good rate of glycemic control compared with NPH insulin-based therapy. Patients initiated with premixed insulin may need strict follow-up and close monitoring. Additionally, patients initiated with a short acting insulin regimen as a component of premixed insulin need to be highly vigilant to recognize the symptoms of hypoglycemia, and self-management to the monitoring of blood glucose levels.

## Supporting information

**S1 File. Dataset.**
(SAV)

## Acknowledgments

The authors would like to thank the data collectors, the hospital administration and medical record unit managers for their positive cooperation during data extraction.

## Author Contributions

**Conceptualization:** Ashenafi Kibret Sendekie.

**Data curation:** Ashenafi Kibret Sendekie, Adeladlew Kassie Netere, Eyayaw Ashete Belachew.

**Formal analysis:** Ashenafi Kibret Sendekie, Adeladlew Kassie Netere, Eyayaw Ashete Belachew.

**Investigation:** Ashenafi Kibret Sendekie, Adeladlew Kassie Netere, Eyayaw Ashete Belachew.

**Methodology:** Ashenafi Kibret Sendekie, Adeladlew Kassie Netere, Eyayaw Ashete Belachew.

**Project administration:** Ashenafi Kibret Sendekie.

**Resources:** Ashenafi Kibret Sendekie.

**Supervision:** Eyayaw Ashete Belachew.

**Validation:** Ashenafi Kibret Sendekie, Adeladlew Kassie Netere.

**Writing – original draft:** Ashenafi Kibret Sendekie.

**Writing – review & editing:** Ashenafi Kibret Sendekie, Adeladlew Kassie Netere, Eyayaw Ashete Belachew.

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
