## [Decision Letter · Decision Letter 0]

22 Aug 2022

PONE-D-22-18276Hypoglycemic events and glycemic control effects between NPH and premixed insulin in patients with type 2 diabetes mellitus: A real-world experience at a comprehensive specialized hospital in EthiopiaPLOS ONE

Dear Dr. Sendekie, 

Thank you for submitting your manuscript to PLOS ONE. After careful consideration, we feel that it has merit but does not fully meet PLOS ONE’s publication criteria as it currently stands. Therefore, we invite you to submit a revised version of the manuscript that addresses the points raised during the review process.

Please submit your revised manuscript by Oct 06 2022 11:59PM. If you need more time than this to complete your revisions, please reply to this message or contact the journal office at plosone@plos.org. Please include the following items when submitting your revised manuscript:A rebuttal letter that responds to each point raised by the academic editor and reviewer(s). You should upload this letter as a separate file labeled 'Response to Reviewers'.A marked-up copy of your manuscript that highlights changes made to the original version. You should upload this as a separate file labeled 'Revised Manuscript with Track Changes'.An unmarked version of your revised paper without tracked changes. You should upload this as a separate file labeled 'Manuscript'.

We look forward to receiving your revised manuscript.

Kind regards,

Rekha Samuel

Academic Editor

PLOS ONE

Journal Requirements:

4. Please include a copy of Table 6 which you refer to in your text on page 12.

**Academic Editor comments: **

Please provide all information to support the methodology of your study. The inclusion of HbA1C as an investigative tool is critical to this study. It is advised that you kindly proof read your manuscript before submission to ensure that there are no grammatical errors

Reviewers' comments:

Reviewer's Responses to Questions

**Comments to the Author**

1. Is the manuscript technically sound, and do the data support the conclusions?

Reviewer #1: No

Reviewer #2: Partly

2. Has the statistical analysis been performed appropriately and rigorously? 

Reviewer #1: I Don't Know

Reviewer #2: Yes

3. Have the authors made all data underlying the findings in their manuscript fully available?

Reviewer #1: No

Reviewer #2: No

4. Is the manuscript presented in an intelligible fashion and written in standard English?

Reviewer #1: No

Reviewer #2: Yes

5. Review Comments to the Author

Reviewer #1: This article has too many grammatical errors. The sentence structure is very poor making it very hard to read and understand. Average fasting glucose is not a very good target marker when comparing diabetic control. The article doesn't make any mention of HbA1C in the two groups.

There are many variables that could affect the outcomes of the NPH group Vs the premixed insulin group results. How was the regimen being followed by the patients standardised? FBG levels taken at random intervals during the year is not sufficient to be a true reflection of the fasting blood glucose for comparison.

Reviewer #2: 

1.Kindly provide information about HbA1C

2.Methodology about hypoglycemia has to be more elaborate

3. You need to provide details of autonomic neuropathy especially hypoglycemia unawareness

4.Please mention units for all variables mentioned in the table

5.Discussion need to be more precise

6. Table 6 information need to be provided

6. PLOS authors have the option to publish the peer review history of their article (what does this mean?). If published, this will include your full peer review and any attached files.

Reviewer #1: No

Reviewer #2: No

---

## [Author Response · Author response to Decision Letter 0]

29 Aug 2022

Responses to the review’s comments

Dear PlOS ONE Academic Editor,

Thank you for giving us the opportunity to submit a revised draft of the manuscript and we would also like to thank your constructive and fruitful comments and suggestions on our paper (Manuscript ID: PONE-D-22-18276). We are very concerned and combined all the suggested comments provided, which we believe that strengthened the paper and we hope this render our paper to be considered for publication in your reputed journal. We appreciate the time and effort that you and the reviewers dedicated to providing feedback on our manuscript and are grateful for the insightful comments on and valuable improvements to this paper.

We authors would like to let you know that all comments and concerns raised by both academic editors and reviewers are fully addressed and indicated with track changes in the main document and a point-by-point response letter for the editors and reviewers. Moreover, we did our best changes and corrections on this revised manuscript. All the changes and corrections are indicated with track changes in the main document. All page and line numbers refer to the revised manuscript file with tracked changes.

Comments from the editor:

#1.... Please ensure that your manuscript meets PLOS ONE's style requirements, including those for file naming. 

Author reply: Thank you for your recommendations to assure adherence to the manuscript template requirements of the journal. Considering to your recommendation, we have adjusted and ensured it accordingly. All files including the main documents, tables and figures are incorporated according to the journal requirements. 

#2…. In your Data Availability statement, you have not specified where the minimal data set underlying the results described in your manuscript can be found. PLOS defines a study's minimal data set as the underlying data used to reach the conclusions drawn in the manuscript and any additional data required to replicate the reported study findings in their entirety. All PLOS journals require that the minimal data set be made fully available.

Author reply: Thank you very much for your valuable important. Considering your comments and suggestions we revised the “Data Availability Statement” as “All relevant data are within the paper and its supporting information (S1 File. Dataset)”. We have also included the data set in the submission documents as “Supplementary files “. The supplementary file information is placed in the the bottom of the manuscript a the end of the reference. 

#3…We note that you have stated that you will provide repository information for your data at acceptance. Should your manuscript be accepted for publication, we will hold it until you provide the relevant accession numbers or DOIs necessary to access your data. If you wish to make changes to your Data Availability statement, please describe these changes in your cover letter and we will update your Data Availability statement to reflect the information you provide.

Author reply: Thank you very much for your valuable information. It was wrongly placed, and we have not changed the data availability statement. 

#4…. Please include a copy of Table 6 which you refer to in your text on page 12.

Author reply: Thank you very much for your kind comments. We had recognized that it was missed after the submission of the manuscript, and although we had asked the editors to send the submission back, but we couldn’t get the chance to provide the missed table. Now, we have provided it. 

Academic Editor comments: 

#1…. Please provide all information to support the methodology of your study. The inclusion of HbA1C as an investigative tool is critical to this study. 

Author reply: Thank very much for your comment to include the HbA1C as a glycemic control measuring variable. Based on the recommendation, we have used HbA1C to measure glycemic control levels of the participants. The glycemic level between NPH and premixed insulin was also compared using HbA1C. FBG was used only to show a change in FBG from baseline after the initiation of either NPH or premixed insulin. 

#2… It is advised that you kindly proof read your manuscript before submission to ensure that there are no grammatical errors.

Author reply: Thank you very much. We had gone through the whole part of the manuscript and we have revised it. We hope you have found that it has improved.

Response to Reviewers’ comments:

Reviewer #1

#1... This article has too many grammatical errors. The sentence structure is very poor making it very hard to read and understand. 

Author response: We authors are very thankful for your deep concerns and suggestions. We, therefore, accepted the recommendations and made editing and corrections for the existing errors and poor sentence structure. We also go through the whole parts of the manuscript to correct the long sentences, redundancy, and existing grammar and writing errors in the manuscript that indicated with track changes. The manuscript changes have tracked and all the raised issues, changes and responses are indicated with those line numbers in the manuscript with tracked changes. We hope that you have found it has improved. 

#2.... Average fasting glucose is not a very good target marker when comparing diabetic control. The article doesn't make any mention of HbA1C in the two groups.

Author reply: Thank very much for your valuable comments regarding to use the HbA1C as a glycemic control measuring variable which improve the quality of this paper. We shared your concerns and comments. Even though there were records of HbA1C, but we used FBG over HbA1c because of inconsistent records and monitoring of patients using HbA1C throughout the follow-up periods. The setting was not use HbA1C for regular monitoring of blood glucose and patients could not monitor using HbA1c every follow-up time they had, but there was a recent record of HbA1C for the participants. However, the participants were monitored and recorded with FBG consistently, which we have found it became easy for us to take a respective FBG records and determining of the glycemic levels to determine glycemic control. However, your concern is clear that determining glycemic control using HbA1c might be valuable over FBG and it estimates the glycemic level of the past months. As a result, based on your valuable comments and suggestions we have revised glycemic outcome measuring variable from FBG to HbA1C (page, lines 22-25). After we have taken the most recent (near to each follow-up periods) HbA1C records of the participants, we determined the level of glycemic control based on ADA recommendations. Generally, based on your recommendation, we have changed glycemic control measuring variable to HbA1C and glycemic control levels of the participants was measured accordingly. The glycemic level between NPH and premixed insulin was also compared using HbA1C. FBG was used only to show a change in FBG from bassline after the initiation of either NPH or premixed insulin. 

#3…There are many variables that could affect the outcomes of the NPH group Vs the premixed insulin group results. How was the regimen being followed by the patients standardized? FBG levels taken at random intervals during the year is not sufficient to be a true reflection of the fasting blood glucose for comparison.

Author reply: Thank you very much for your valuable comments which improve the quality of this paper. We shared your concern and comments. Yes, there are variables that could affect the outcomes. Patients were initially initiated with two different insulin regimens after they have failed to achieve target glycemic levels with initial therapy. In this study, the participants were assigned to NPH vs premixed insulin regimens at the indexed time (the time at which insulin NPH or premixed) and followed to the end of follow-up period. Patients with an altered insulin type (NPH to premixed or viscera) during a one-year follow-up period were excluded from the outset at indexed time. Insulin initiation and treatment intensification as well as dose titrations have to be based on the ADA recommendations. But we just followed the medical records of the participants retrospectively. (This section is revised on page 6, lines 7-12)

In this regard, there may be a lot of confounders potentially altered the outcomes of NPH vs Premixed insulin groups. But we assessed the association of potential variables with the study variables (hypoglycemia and glycemic control) using adjusted multivariable logistic regression analysis. As a result, besides NPH vs Premixed insulin regimens, we have found some variables which could have an effect with the outcomes (hypoglycemia and glycemic control).

Regarding to FBG, based on your valuable comments and suggestions, we have revised the outcome measuring variable to HbA1C (page 7, lines 22-25). By taking the most recent HbA1C record to the follow-up periods, we measured the glycemic control levels using the average HbA1C level. Then, the outcomes (hypoglycemia and glycemic control) were compared for the treatment regimens based on our outcome measures. We also examined the association of other variables affects hypoglycemia and glycemic control. 

Reviewer #2

#1… Kindly provide information about HbA1C

Author reply: Thank very much for your comment to provide HbA1C as a glycemic control measuring variable which improve the quality of the paper. Even though there were records of HbA1C, but we used FBG over HbA1c because of inconsistent records and monitoring of patients using HbA1C throughout the follow-up periods. The setting was not use HbA1C for regular monitoring of blood glucose and patients could not monitor using HbA1c every follow-up time they had, but there was a recent record of HbA1C for the participants. However, the participants were monitored and recorded with FBG consistently, which we have found it became easy for us to take a respective FBG records and determining of the glycemic levels to determine glycemic control. However, your concern is clear that determining glycemic control using HbA1c might be valuable over FBG and it estimates the glycemic level of the past months. As a result, based on your valuable comments and suggestions we have revised glycemic outcome measuring variable from FBG to HbA1C (page 7, lines 22-25). After we have taken the most recent (near to each follow-up periods) HbA1C records of the participants, we determined the level of glycemic control based on ADA recommendations. Generally, based on your recommendation, we have changed glycemic control measuring variable to HbA1C and glycemic control levels of the participants was measured accordingly. The glycemic level between NPH and premixed insulin was also compared using HbA1C. FBG was used only to show a change in FBG from bassline after the initiation of either NPH or premixed insulin. 

Methodology:

#2… Methodology about hypoglycemia has to be more elaborate.

Author reply: Thank you very much for your important comments and suggestions. We shared your concern regarding hypoglycemia assessment and reporting. Based on your comments we tried to elaborate and revised the methodology about hypoglycemia (page 7, lines 13-21). As we mentioned this study is a retrospective follow-up done on medical records of two insulin regimens treated patients with T2DM. Therefore, we assessed the rate of hypoglycemia among the patients using available relevant witness. Because of this we could not include direct patients’ experience and reports regarding hypoglycemic events in the study methodology. For this, we followed a one-year record of patients and find out the recorded clinical episodes of hypoglycemia which patients experience when their blood glucose levels become below normal such as sweating, shaking, fatigue, fainting, dizziness, confusion, and/or a blood glucose level recorded in the range of hypoglycemia (< 70 mg/dl). Patients experienced at least one episode of hypoglycemia or a record of blood glucose level as considered to have hypoglycemia. Finally, it is measured and reported based on the number of episodes per patients during the follow-up period. 

#3… You need to provide details of autonomic neuropathy especially hypoglycemia unawareness

Author reply: Thank you very much to your comments and suggestions to provide information about hypoglycemia unwariness to improve the quality of the paper. Considering your important suggestions and comments we provide short description regarding HU on the method section under operational definition (page 6, lines 14-17). 

#4…. Please mention units for all variables mentioned in the table

Author reply: Thank you for your comments and suggestions. Taking your valuable comments, we have included the units of the mentioned variables. 

#5… Discussion needs to be more precise

Author reply: We authors are very grateful for the concerns and recommendations you raised to concise the discussion. Considering your valuable comments and suggestions, we revised the discussion section and have revised and tried to concise. The changes are found with tracks in the manuscript with track changes. We hope, you have found that it has improved.

#6… Table 6 information need to be provided

Author reply: Thank you very much for your kind comments. We had recognized that it was missed after the submission of the manuscript, and although we had asked the editors to send the submission back, but we couldn’t get the chance to provide the missed table. Now, we have provided it (page 12).

---

## [Editor Report · Decision Letter 1]

9 Sep 2022

Hypoglycemic events and glycemic control effects between NPH and premixed insulin in patients with type 2 diabetes mellitus: A real-world experience at a comprehensive specialized hospital in Ethiopia

PONE-D-22-18276R1

Dear Dr. Sendekie,

We’re pleased to inform you that your manuscript has been judged scientifically suitable for publication and will be formally accepted for publication once it meets all outstanding technical requirements.

Kind regards,

Rekha Samuel

Academic Editor

PLOS ONE

---

## [Editor Report · Acceptance letter]

12 Sep 2022

PONE-D-22-18276R1 

Hypoglycemic events and glycemic control effects between NPH and premixed insulin in patients with type 2 diabetes mellitus: A real-world experience at a comprehensive specialized hospital in Ethiopia    

Dear Dr. Sendekie:

I'm pleased to inform you that your manuscript has been deemed suitable for publication in PLOS ONE. Congratulations! Your manuscript is now with our production department. 

Kind regards, 

on behalf of

Dr. Rekha Samuel 

Academic Editor

PLOS ONE